# Starch Retrogradation in Rice Cake: Influences of Sucrose Stearate and Glycerol

**DOI:** 10.3390/foods9121737

**Published:** 2020-11-25

**Authors:** Seon-Min Oh, Hee-Don Choi, Hyun-Wook Choi, Moo-Yeol Baik

**Affiliations:** 1Department of Food Science and Biotechnology, Institute of Life Science and Resources, Kyung Hee University, Seochun 1, Yongin 446-701, Korea; seonminoh@khu.ac.kr; 2Korean Food Research Institute, 245, Nongsaengmyeong-ro, Iseo-myeon, Wanju-gun, Jeollabuk-do 55365, Korea; chdon@kfri.re.kr; 3Department of Functional Food and Biotechnology, College of Medical Sciences, Jeonju University, 303 Cheonjam-ro, Jeonju 55069, Korea; hwchoi96@jj.re.kr

**Keywords:** rice cake, glycerol, sucrose fatty acid ester, starch retrogradation, kinetic

## Abstract

Retrogradation properties and kinetics of rice cakes with the addition of glycerol (GLY) and sucrose fatty acid ester (SE) were investigated. In hardness, both rice cakes with glycerol (RGLY) and rice cakes with sucrose fatty acid ester (RSE) showed lower initial hardening compared with the control for up to 5 days. X-ray diffraction (XRD) pattern of RSE showed a B+V-type pattern, and the relative crystallinity showed that GLY and SE lowered the initial and final crystallization of rice cake. Both GLY and SE affected the retrogradation enthalpy, glass transition temperature, and ice melting enthalpy in differential scanning calorimeter (DSC). However, ^1^H NMR relaxation time (T_2_) of rice cake decreased regardless of additives. From these results, the addition of glycerol and sucrose stearate inhibits the retrogradation process of rice cakes, which will solve industrial problems. Applying the Avrami equation for retrogradation kinetics of rice cake was suitable in XRD and DSC with high coefficient of determination (0.9 < *R*^2^). Meanwhile, the other retrogradation index improved the *R*^2^ when the exponential rise to maximum equation was used. This suggests that there is an alternative of Avrami equation to predict the retrogradation.

## 1. Introduction

Rice cake is a traditional food developed since ancient time and mainly consumed in Asian countries. It is made with rice flour and water, and sugar or salt is also added according to preference. Fresh rice cake has a sticky and soft texture, however, it tends to harden easily, leading to poor digestibility and quality during storage. This is because starch, which is the main component of rice cake, will reassociate and recrystallize over time [1,2]. This phenomenon shortens the shelf-life of rice cakes and limits their distribution [3,4]. Therefore, many studies have attempted to retard starch retrogradation of rice cakes using various additives [5].

Sucrose fatty acid esters (SEs) are known as sugar esters, and non-ionic-type emulsifiers [6]. Because sucrose has eight free hydroxyl groups, SE can exist in the form of monoesters to octaesters depending on the degree of esterification with a fatty acid. SEs consist of a hydrophilic sugar head and one or more lipophilic fatty acids. Mono- or poly-esterified SEs have a hydrophilic–lipophilic balance (HLB) value from 0 to 16, and the hydrophilicity tends to get higher as the ratio of monoester becomes high [7]. SE has been reported to retard or inhibit starch retrogradation by forming a complex with hydrophobic region in the amylose [8]. SE with an HLB value of 11, which has an equal balance of hydrophilic and lipophilic regions, has shown a low firming rate in rice starch gel [9]. Meng et al. [6] reported that the SE with an HLB value of 15 improved the freeze–thaw stability by preventing the re-association of starch chains.

Glycerol is often used as a plasticizer to prepare thermoplastic starch as well as to improve the flexibility and processability of starch. However, it has been reported to exhibit either plasticizing or antiplasticizing effects depending on the presence of other plasticizer and its specific concentration ranges. These conflicting properties significantly affect the glass transition, crystallization, plasticizing, and physical properties of a starchy product [10]. Baik and Chinachoti [11] reported that three –OH groups influenced the networking and staling rate of bread by osmotic hydration. Meanwhile, when glycerol was added to a waxy maize starch–water mixture, retrogradation was reduced as glycerol content increased because of increased interactions between starch–glycerol or glycerol–water [12].

SE and glycerol are representative chemical additives known as inhibitors of starch retrogradation. As an emulsifier or a plasticizer, their mode of action and mechanism of retarding starch retrogradation are different. There is limited information on the effects of these two materials on the starch retrogradation in terms of their roles in crystalline and amorphous regions. Furthermore, there is also a lack of research on macro-, micro-, and molecular level studies on the effects of their roles in starch retrogradation kinetics. Therefore, this study investigated the retrogradation properties and kinetics of rice cakes with glycerol or SE on the macro- (hardness), micro- (DSC, XRD), and molecular levels (NMR). On the basis of these results, the retrogradation kinetics of rice cakes were analyzed and the suitability of the Avrami equation was determined in each level.

## 2. Materials and Methods

### 2.1. Materials

Wet-milled normal rice flour (33.9% moisture content) was purchased from Hanwool-Food (Hwaseong, Korea). Sucrose stearic acid esters (S-1670, Mitsubishi-Kagaku Foods Co., Tokyo, Japan) with HLB values of 16 and food grade glycerol (LG Household & Health Care, Ltd., Seoul, Korea) with 99% purity were used.

### 2.2. Preparation of Rice Cake

After preparing the wet-milled normal rice flour–water mixture with 45% moisture content, glycerol (GLY) and SE were added. Through a preliminary experiment, the concentration of the additive was determined in a range possessing a texture similar to that of a commercial rice cake. The GLY (1%, 5%, and 10%/g rice flour) and SE (0.1%, 0.3%, and 0.5%/g rice flour) were added to rice flour–water mixture. The mixture was steamed using boiling water for 20 min to be fully gelatinized, and then a rice cake was prepared using a noodle maker (MS-30000, Oscar Electronic Co., Seoul, Korea) with round-shaped nozzle of 1.8 cm diameter. The size of prepared rice cake was 1.8 cm (diameter) × 10 cm (length). Depending on the amount of additive, rice cakes with GLY were described as RGLY1, RGLY5, and RGLY10, and rice cakes with SE were described as SE0.1, SE0.3, and SE0.5. Rice cake without any additive was referred to as control. The rice cake was allowed to stand for 15 min at room temperature (25 ± 1 °C) to remove the moisture on the surface. Rice cakes were vacuum-packaged in a multilayer film (Freshield, CSE, Siheung, Korea) and stored for 0, 1, 2, 3, 4, 5, 9, and 14 days at 25 °C.

### 2.3. Hardness

Cylindrical-shaped samples (1.8 cm diameter × 2 cm length) were used for hardness measurement. The specimens were placed between parallel plates fitted to a rheometer (CR-200D; Sun Scientific, Tokyo, Japan). Using flat probe (*d* = 24 mm), the hardness was assessed under a fixed cross-head speed of 300 mm/min and chart speed of 300 mm/min. The rice cake was compressed once to 60% of the initial height, and the height of the first peak on the curve was taken as the hardness (N).

### 2.4. X-ray diffraction Pattern

X-ray diffraction patterns and relative crystallinity of rice cakes were obtained using a D8 Advance X-ray diffractometer (Bruker Co., Karlsruhe, Germany) with the 40 kV voltage and 40 mA current. To dehydrate the moisture of the rice cake and terminate the retrogradation process, the rice cake was ground with 99% ethanol and dried in an oven at 40 °C, and filtered through an 80-mesh. The X-ray diffraction pattern of dried rice cake powder was recorded from 3–40° of the diffraction angle (2*θ*) at a rate of 6°/min. The relative crystallinity was calculated as the ratio of the crystalline peak area to the total diffraction area using Sigma Plot software (version 6.0 Jandel Scientific, San Rafael, CA, USA) [13].

### 2.5. DSC Thermal Properties

Thermal properties were determined using a differential scanning calorimeter (DSC 4000, PerkinElmer, Waltham, MA, USA). After removing the outer shell, the center portion of the rice cake (10 mg) was taken and put into an aluminum pan and hermetically sealed. The sample was cooled down to −50 °C using liquid nitrogen and equilibrated for 30 min. It was then heated from −50 to 120 °C at a heating rate of 5 °C/min. An empty pan was used as a reference. DSC thermogram was analyzed using Pyris software (version 11.1.0.04.88., Perkin Elmer., Inc., Waltham, MA, USA). Based on the method of Baik et al. [12], the glass transition temperature (T_g_’) of rice cake was determined as the end point from the first derivative curve of the DSC thermogram. The changes in T_g_’, ice melting enthalpy (∆H_i_), and amylopectin melting enthalpy (∆H_r_) of rice cakes were assessed [14,15].

### 2.6. H NMR Analysis 

Proton relaxation measurement of retrograded rice cake was carried out using a low-resolution (20 MHz) ^1^H NMR spectrometer (Minispec mq pulsed NMR, Bruker, Whitney, UK). The samples were placed in 16 mm diameter NMR glass tubes and covered with a cap to avoid moisture loss. All measurements were performed at 40 °C. The Carr–Purcell–Meiboom–Gill (CPMG) pulse sequence was used for acquisition of the free induction decay (FID) data for the transverse relaxation time (T_2_). The experimental parameters were set appropriately to maximize the signal-to-noise ratio and to cover the entire relaxation range as completely as possible. The FID(s) obtained from the CPMG pulse sequence were analyzed using a mono-exponential model: 90x − (t − 180y − τ − echo)*n*. Pulse lengths with 90° and 180° pulses were separated by 20 μs, a relaxation delay of 6 s, and a τ of 0.7, and 120 data points measurements were performed using RI WinDXP software (version 1.2.2., Resonance Instruments Ltd., Oxfordshire, UK).

### 2.7. Retrogradation Kinetics 

The following Avrami equation was applied to determine the retrogradation rate of rice cakes; [16].
(1)$$\theta =\left(E_{L}-E_{t}\right)/\left(E_{L}-E_{0}\right)=\exp\left(-kt^{n}\right)$$
(2)$$\log\left(-\ln\;\left(E_{L}-E_{t}\right)/\left(E_{L}-E_{0}\right)\right)=\log k+n\log t$$
where:*θ* is the region of nonretrograded material remaining after time *t*. *E_L_* is the maximum value of hardness, relative crystallinity, DSC results (T_g_’, ΔH_i_ and ΔH_r_), and T_2_ from NMR result. *E_t_* indicates the value of hardness, relative crystallinity, DSC results (T_g_’, ΔH_i_ and ΔH_r_), and T_2_ from NMR result at time *t*.*E*_0_ represents the value of hardness, relative crystallinity, DSC results (T_g_’, ΔH_i_ and ΔH_r_), and T_2_ from NMR result at time 0. *k* is a rate constant.*n* is the Avrami exponent.

### 2.8. Statistical Analysis

All experiments were performed at least in triplicate. Experimental data were examined using analysis of variance (ANOVA) and expressed as mean ± standard deviation. All statistical computations and analyses were performed using SAS software (version 8.02; SAS Institute, Inc., Cary, NC, USA).

## 3. Results and Discussion

### 3.1. Hardness

The initial hardness of control increased rapidly and approached maximum hardness at day 2 (Figure 1). On the other hand, when GLY (Figure 1A) or SE (Figure 1B) was added, the hardness increased slowly and approached maximum hardness at days 4 and 5 depending on the concentration of the additive. The firming rate of rice cakes decreased with increasing concentrations of both GLY and SE. This result suggested that the GLY and SE decreased the initial firming rate of rice cake but did not lower the maximum hardness. Hardness is a typical macroscopic property in determining starch retrogradation as it is apparent and can be easily recognized. It is a factor that is directly related to starch retrogradation and increases proportionally to the degree of the retrogradation of stored rice cake [17]. Generally, the quality of rice cakes decreases rapidly, mainly due to starch retrogradation, and they cannot be sold after 24 h of showcase storage at room temperature [5]. Therefore, retarding firming rate by adding GLY or SE would be a suitable method to extend the shelf-life of rice cake.

As a plasticizer, glycerol easily penetrates to the amorphous region due to its low molecular weight and reduces the firmness of starchy foods by blocking the aggregation of starch chains or by providing mobility for molecules to move within the polymer structure [11,18]. Moreover, polyols like glycerol improve the water-holding capacity in bread dough and increase the softness of bread. However, the high concentration of glycerol (≥10%) can also increase the hardness of bread by plasticizing and strengthening the starch and gluten [19]. 

The addition of SE also showed great effectiveness in retarding the firming rate of rice cake. SE has been reported to significantly decrease the degree of retrogradation of rice flour gel [20]. SE’s contribution to the cross-linking effect increases the freeze–thaw stability and storage modulus of rice gel, helping retard long-term starch retrogradation [6]. Many studies explained the retrogradation–inhibition effect of emulsifiers, including SE, as the interaction between emulsifiers and amylose [21]. Mun et al. [20] reported that the emulsifier binds with not only amylose but also amylopectin, forming a complex to change the distribution of moisture in the starch, thereby inhibiting retrogradation. The formation of amylose–lipid complexes was also found in our XRD results (Section 3.2), and the interaction between SE and amylopectin molecules cannot be ruled out because the rice cake system utilized in this study also contains amylopectin.

### 3.2. X-ray Diffraction Pattern

The XRD patterns of rice cake with glycerol (RGLY) or sucrose fatty acid ester (RSE) at different concentrations are shown in Figure 2A. The XRD pattern of starch reflects how the molecules are arranged and is classified into three types (A, B, and C) according to the starch sources [2]. Native rice flour revealed a typical A-type crystal pattern with peaks at 15°, 17°, 18°, and 23° (2*θ*). The fresh control showed an overall amorphous pattern, but small peaks were detected at 13°and 20° (2*θ*). This is possibly due to formation of amylose–lipid as well as amylopectin–lipid complexes and partial recrystallization. The retrograded rice cake showed peaks at 5.4°, 17°, and 20° (2*θ*), which is a typical B-type crystal pattern. The peaks at 17° and 20° became sharper and larger in retrograded rice cake, indicating an increase of crystal formation [22,23]. 

After 14 days of storage, RGLY and RSE showed relatively lower peak intensity in comparison with control, implying that the addition of glycerol and SE reduced the recrystallization of rice cake by introducing it into the starch. Interestingly, RSE revealed a broad shoulder around 13° (2*θ*), which is a typical V-type crystal pattern [21]. The results suggest that the hydrophilic head or the hydrophobic tail of SE interacted with the starch and formed amylose–SE or amylopectin–SE complexes. Lipids can induce the formation of complexes by becoming entrapped within and weakening the helical structure of amylose. This amylose–lipid complex has been reported to retard starch recrystallization by interfering with the formation of stable double helical structure [24]. Previous literature also has proposed that lipids could form complexes with outer branches of amylopectin [2].

Figure 2B,C showed the relative crystallinity change of RGLY and RSE during storage. The relative crystallinity of rice cake was greatly influenced by type and concentration of additive. GLY lowered the overall relative crystallinity of rice cake (Figure 2B). On the other hand, SE did not affect the relative crystallinity of rice cake for up to 5 days and lowered it after then (Figure 2C). Both GLY and SE interacted with starch and water molecules and retarded the recrystallization in different ways. GLY acts as a plasticizer, which interferes with the hydrogen bonding between starch and water molecules, resulting in an overall inhibition of recrystallization. In the case of SE, it appears that the inhibition of recrystallization is prominent in the later period as the amylose or amylopectin–lipid complex gradually forms. As shown in Figure 2A, RSE clearly formed the amylose–lipid or amylopectin–lipid complex during storage, consequently retarding the recrystallization of rice cake.

Similar to the result of hardness (Figure 1), GLY and SE were notably effective in inhibiting the recrystallization of starch in rice cake. However, the decrease in hardness was effective in initial storage (within 5 days), and relative crystallinity decreased at later storage (after 5 days). It indicated that retrogradation of rice cake in macro level (hardness) and micro level (XRD) was not always concurrent.

### 3.3. DSC Thermal Properties

In this study, changes in glass transition temperature (T_g_’) and ice melting enthalpy (ΔH_i_) as well as amylopectin melting enthalpy (ΔH_r_) were determined to investigate the status of water and starch molecules in amorphous and crystalline regions. While GLY greatly changed T_g_’ (Figure 3A), SE did not show any effect on the T_g_’ of rice cake (Figure 3D). As a second-order transition in an amorphous region, the glass transition is an important factor that controls the crystallization of the amorphous region, and it depends on molecular characteristic, free volume, and molecular weight [25,26]. T_g_’ of RGLY1 (−5.45 °C), RGLY5 (−8.71 °C), and RGLY10 (−13.19 °C) was lower than that of control (−3.38 °C) at day 0, suggesting that glycerol had a plasticizing effect in this rice cake system. Previous literature has shown that a plasticizer increases the free volume of starch, thereby reducing the glass transition temperature [27]. Consequently, glycerol promotes the mobility of polymer chains and increases the free volume, lowering the glass transition temperature. In addition, the synergistic effect of glycerol and water decreased the T_g_’ dramatically in this system. Although glycerol lowered the initial T_g_’ of rice cake, it did not prevent the increase of T_g_’ during retrogradation, and even a dramatic increase in T_g_’ was observed in RGLY10, possibly due to the increased interaction between glycerol and water during retrogradation. Glycerol has been reported to exhibit the plasticizing or antiplasticizing effect in two or more plasticizer systems [11]. In a fresh state, glycerol plays a plasticizer role and mainly interacts with starch. During retrogradation, glycerol acts as an antiplasticizer and mainly interacts with water due to its hygroscopicity. Consequently, it mainly interacts with water molecules and reduces the plasticizing effect of water molecules, resulting in a dramatic increase in T_g_’ during retrogradation of rice cake. On the other hand, since the glass transition temperatures were at the sub-zero range, it is possible to attribute the increase in T_g_’ during storage to the increased rigidity of the unfrozen phase, which could have undergone substantial molecular associations.

Changes in ΔH_i_ of RGLY and RSE during storage are shown in Figure 3B,E, respectively. In all samples, ΔH_i_ tended to decrease with increasing storage time, as opposed to T_g_’. ΔH_i_ designates the amount of freezable water in the system. Thus, the decrease in ΔH_i_ implied that freezable water becomes unfreezable water or bound water. Gelatinized starch undergoes the structural rearrangement during retrogradation from amorphous starch molecules to B-type crystalline structure, which contains 36 water molecules per 12 glucose residues [28]. Thus, free water molecules entered the starch crystalline lattices, incorporated into them, and changed to bound water molecules, reducing the amount of freezable water [14,29]. Different initial ΔH_i_ of rice cake comes from the varied interactions between additives and water molecules. The control rice cake showed a gradual decrease in ΔH_i_ during retrogradation, but ΔH_i_ of both RGLY and RSE gradually decreased during the first 5 days of storage and then stabilized thereafter. This result suggests that less freezable water changed to unfreezable water under presence of GLY and SE in this system. 

The addition of GLY and SE decreased the ΔH_r_ of rice cake (Figure 3C,F). In general, ΔH_r_ designates the amount of produced amylopectin double helical structure in both amorphous and crystalline regions during retrogradation. It has been reported that polyols, such as glycerol, retard starch retrogradation [30,31]. The high-moisture-retention ability of glycerol contributed to water molecules not being incorporated into the crystalline lattices, retarding the starch retrogradation [32]. In addition, glycerol was introduced into the amorphous region of starch, causing steric hindrance of molecules and preventing the recrystallization of starch [11]. In the case of SE, several hydroxyl groups in SE can form a broad hydrogen-bonding network with starch molecules and water molecules. This reduces the mobility of the starch chain and delays the recrystallization of amylopectin by increasing entanglement or cross-linking between starch molecules [6]. The emulsifier, such as SE, adheres to the surface of the starch molecules in amylopectin, changing the distribution of water and interaction with the side chain through hydrogen bonding. It has been reported that 0.2% of SE is effective in the retardation of starch retrogradation [31], whereas in this study, RSE0.1 did not retard the initial amylopectin recrystallization, indicating that a certain amount of emulsifier is necessary to retard starch retrogradation.

### 3.4. Solid-State ^1^H NMR Transverse Relaxation Time (T_2_)

Figure 4 presents the changes in ^1^H NMR transverse relaxation time (T_2_) of RGLY and RSE during storage. The low-magnetic-field ^1^H NMR is the most common NMR technique to investigate the mobility of protons. ^1^H NMR T_2_ is known as spin–spin relaxation time or transverse relaxation time, which have long or short relaxation time depending on the immobile/mobile state of the proton [2]. The ^1^H NMR T_2_ values of all samples decreased with increasing storage time. This indicated that proton mobility in the rice cake changed from a more mobile state to a less mobile state. As retrogradation proceeds, the disordered state of starch transforms to an ordered state, and water molecules are incorporated into the crystalline lattices, resulting in a decreased mobility of water [33]. It has been reported that pasta containing glycerol (5% and 15% of substitution) has high water mobility because the plasticizing effect of glycerol in pasta matrix increases the flexibility of water molecules [34]. Moreover, the mobility of water decreases or increases depending on the amount of SE in the fresh state, and the T_2_ decreases gradually during the retrogradation [21]. As such, the decrease in ^1^H NMR T_2_ during starch retrogradation is a well-known phenomenon. However, in this study, the addition of GLY slightly increased ^1^H NMR T_2_ values in the fresh rice cake, possibly due to the plasticizing effect of glycerol. Similarly, SE also changed the initial ^1^H NMR T_2_ values of rice cake. However, the addition of GLY or SE did not retard the decrease in ^1^H NMR T_2_ of rice cake during storage. Overall, the addition of GLY and SE was effective in initial ^1^H NMR T_2_ values, but their amounts were not enough to retard the decrease in ^1^H NMR T_2_ values during retrogradation.

### 3.5. Retrogradation Kinetics

Retrogradation kinetics of rice cakes were investigated using the determined retrogradation properties. The Avrami equation is generally used to investigate the retrogradation kinetics of starch. The retrogradation rate constant (*k*), Avrami exponent (*n*), and coefficient of determination (*R*^2^) can be obtained from the Avrami equation. Before determining the *k* or *n*, the results obtained from macro-, micro-, and molecular-level retrogradation properties were assessed to determine whether the Avrami equation could be applied to the results. Figure 5 shows the result of substituting the Equation (1) in Section 2.7, and the *R*^2^ values are shown in Table 1. The classic retrogradation index, such as XRD and ΔH_r_, is well suited to the Avrami equation for measuring the retrogradation kinetics of starch with relatively high *R*^2^ values from 0.94 to 0.99. Hardness, T_g_’, and ΔH_i_ showed a relatively wider range of *R*^2^ from 0.81 to 0.96, suggesting that fitting the results to the Avrami equation is possible in this case, because they followed the classic retrogradation index. On the other hand, the T_2_ revealed a lower and wider range of *R*^2^ than that of classic retrogradation index, from 0.48 to 0.82. This indicates that the determination of retrogradation kinetics using the Avrami equation is not suitable in this case. In the case in which the application of the Avrami equation is inappropriate, empirical modelling can be used to deduct the retrogradation kinetics of starch in complex starch systems [35].

Therefore, retrogradation kinetics of the assessed retrogradation properties excluding ΔH_r_ and XRD were additionally investigated using a nonlinear regression analysis. The following exponential rise to maximum equation that is built-in the sigma library was applied.
(3)$$y=y_{0}+a\left(1-\;e^{-bx}\right)$$

*x* is an independent variable assigned to storage time *t,* and y is a dependent variable assigned to hardness, T_g_’, ∆H_i_ and T_2_, respectively. To find the values of $y_{0}$, *a,* and *b*, the software automatically implemented regression. Therefore, this equation describes the data (*P* < 0.0001) by using the values of the independent variables to predict the values of the dependent variables.

Figure 6 shows the curve fitting of hardness, ΔH_i_, T_g_’, and T_2_ using the exponential rise to maximum equation, respectively, and their *R*^2^ values are shown in Table 1. When applied to the exponential rise to maximum equation, the *R*^2^ values increased in all cases and were in the range of 0.85–0.99. This result suggests that the exponential rise to maximum equation can be an alternative to determine retrogradation kinetics. Previous work has confirmed that DSC and NMR data of some samples did not fit well with the Avrami equation, but showed high correlation coefficient (*R*^2^ > 0.99) using nonlinear regression analysis [35]. Therefore, the retrogradation kinetics of starch can be determined by the Avrami equation preferentially, and the application of other nonlinear regression equations may be an alternative if the Avrami equation is not suitable.

## 4. Conclusions

In this study, retrogradation properties of rice cake with glycerol or sucrose fatty acid ester were investigated at macroscopic (Hardness), microscopic (DSC and XRD), and molecular levels (^1^H NMR). When glycerol and sucrose fatty acid ester were added to rice cake, they interfered with the starch re-association and recrystallization. Glycerol acted as a plasticizer, interacted with water and starch molecules, improved the water-holding capacity, and inhibited the starch retrogradation in macroscopic and microscopic levels. In case of adding SE, a V-type pattern was obtained in XRD and this can be attributed to the formation of amylose– or amylopectin–lipid complexes. Through these results, both substances were shown to be consequently effective in the retardation of starch retrogradation through differing modes of action. Starch retrogradation is a complex and complicated phenomenon that cannot be justified by a single investigation. Therefore, three levels of investigation would be helpful to understand the starch retrogradation mechanism. Additionally, the Avrami equation can be generally applied to the classic retrogradation properties to determine the starch retrogradation kinetics. However, there is a limit in applying the Avrami equation in certain determinations because starch retrogradation kinetics are complex. Therefore, applying other models for specific results, instead of the Avrami equation, can be an alternative to analyze reliable starch retrogradation kinetics.

## Figures and Tables

**Figure 1 foods-09-01737-f001:**
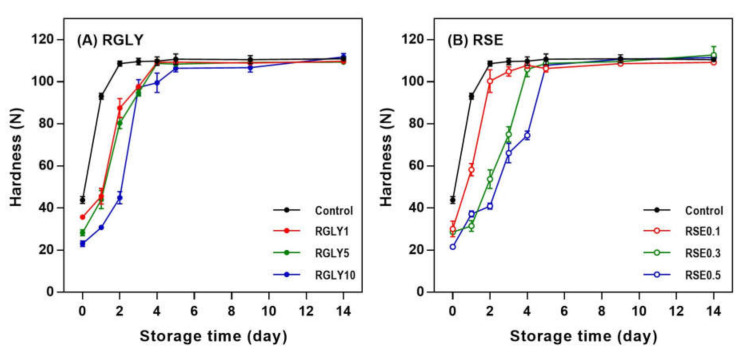
Changes in hardness of rice cakes with glycerol (**A**) or sucrose fatty acid ester (**B**) during retrogradation. RGLY: rice cakes with glycerol; RSE: rice cakes with sucrose fatty acid ester.

**Figure 2 foods-09-01737-f002:**
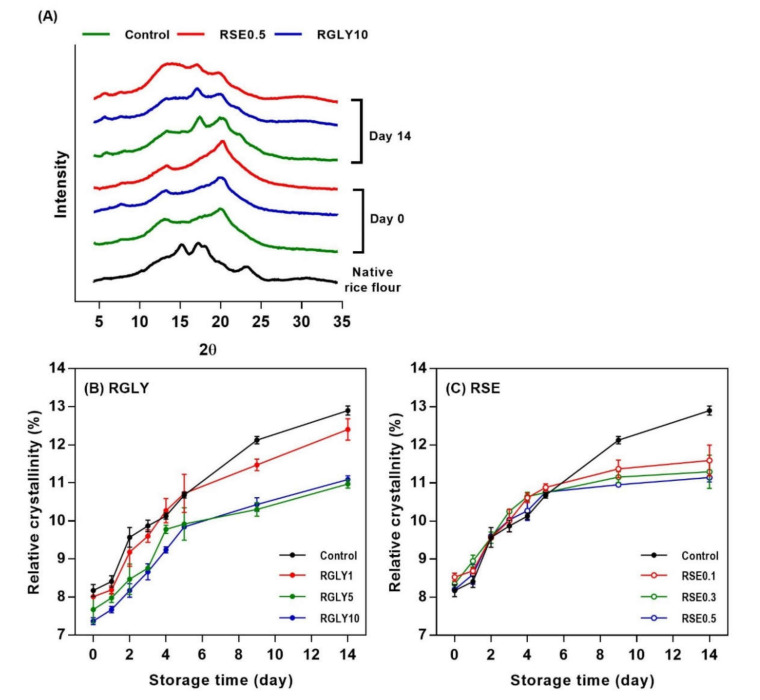
X-ray diffraction patterns of fresh and retrograded rice cakes (**A**), and changes in relative crystallinity of rice cakes with glycerol (**B**) and sucrose fatty acid ester (**C**) during retrogradation.

**Figure 3 foods-09-01737-f003:**
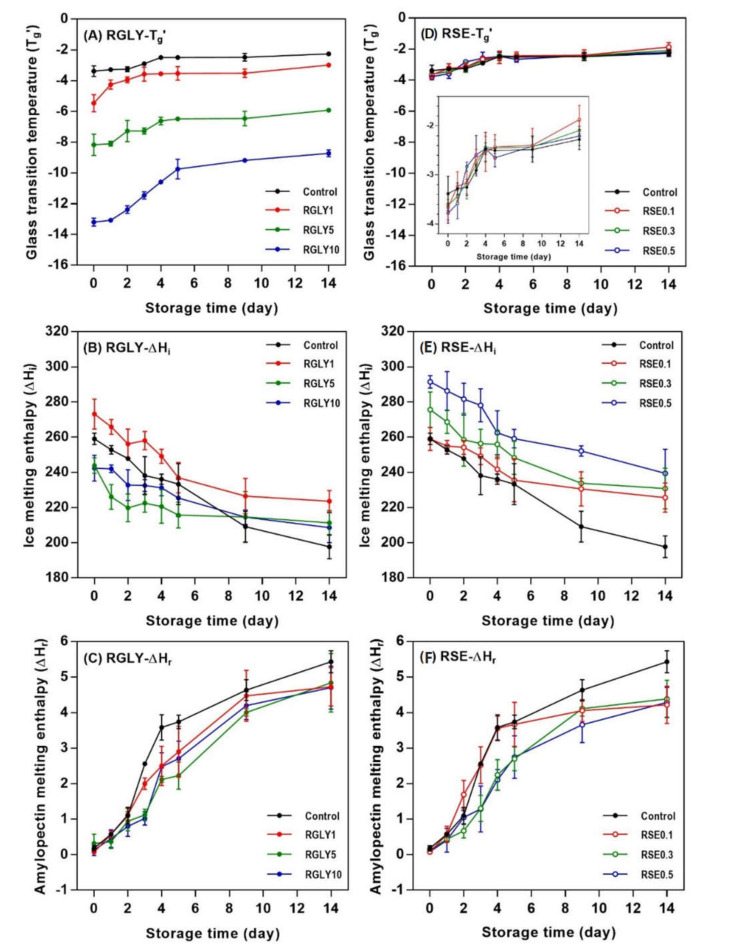
Changes in glass transition temperature (T_g_’), ice melting enthalpy (ΔH_i_), and amylopectin melting enthalpy (ΔH_r_) of rice cakes with glycerol (**A**–**C**) and sucrose fatty acid ester (**D**–**F**) during retrogradation, respectively.

**Figure 4 foods-09-01737-f004:**
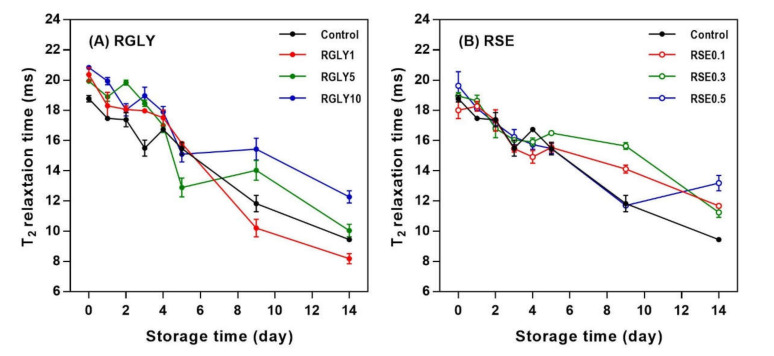
Change of ^1^H NMR transverse relaxation time (T_2_) of rice cakes with glycerol (**A**) and sucrose fatty acid ester (**B**) during retrogradation.

**Figure 5 foods-09-01737-f005:**
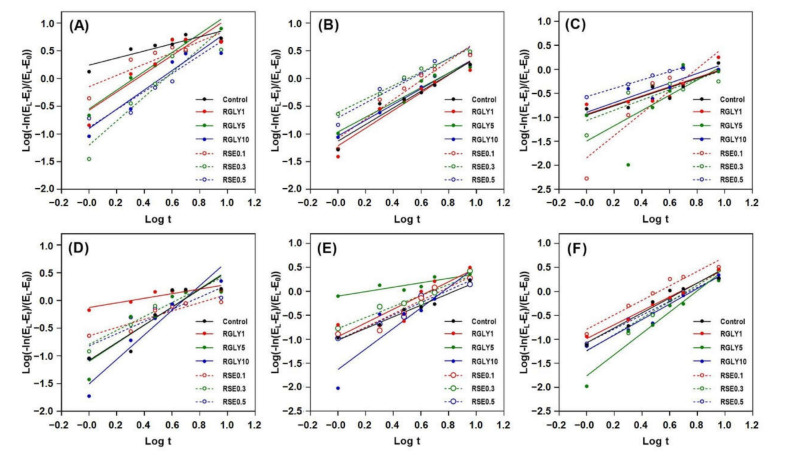
Retrogradation kinetics of rice cakes with glycerol or sucrose fatty acid ester using Avrami equation; hardness (**A**), relative crystallinity (**B**), ^1^H NMR transverse relaxation time (**C**), glass transition temperature (**D**), ice melting enthalpy (**E**) and amylopectin melting enthalpy (**F**).

**Figure 6 foods-09-01737-f006:**
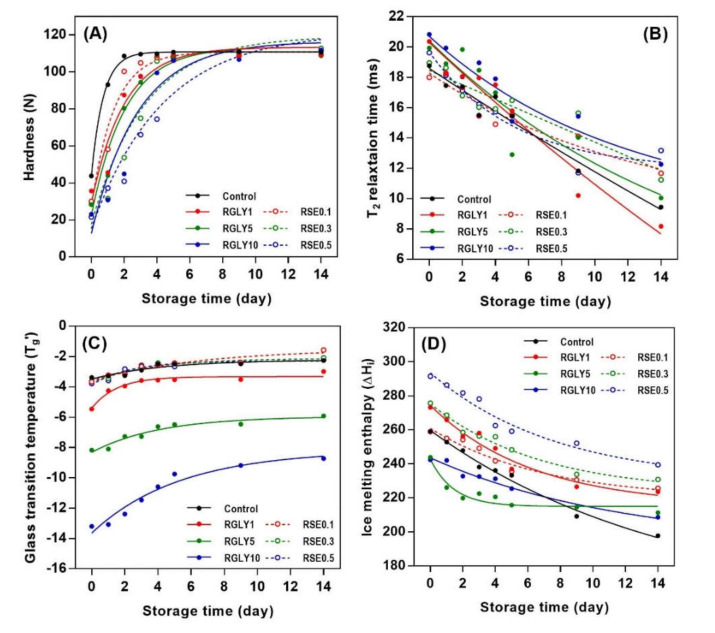
Retrogradation kinetics of rice cakes with glycerol or sucrose fatty acid ester using exponential rise to maximum equation; hardness (**A**), ^1^H NMR transverse relaxation time (**B**), glass transition temperature (**C**), and ice melting enthalpy (**D**).

**Table 1 foods-09-01737-t001:** Coefficient of determination (*R*^2^) of retrogradation kinetic analysis using Avrami and exponential rise to maximum equations.

		Coefficient of Determination (*R*^2^)						
		Control	RGLY1	RGLY5	RGLY10	RES0.1	RSE0.3	RSE0.5
Avrami	Hardness	0.81	0.83	0.94	0.85	0.81	0.89	0.90
	XRD	0.94	0.93	0.96	0.99	0.94	0.98	0.94
	T_g_’	0.83	0.83	0.81	0.94	0.85	0.87	0.82
	ΔH_i_	0.96	0.81	0.79	0.84	0.91	0.93	0.94
	ΔH_r_	0.95	0.99	0.95	0.95	0.96	0.97	0.98
	T_2_	0.78	0.71	0.48	0.79	0.82	0.69	N.D.
Exponential rise to maximum	Hardness	0.99	0.93	0.95	0.88	0.95	0.91	0.92
	T_2_	0.96	0.95	0.85	0.90	0.92	0.86	0.92
	T_g_’	0.89	0.95	0.94	0.95	0.92	0.94	0.92
	ΔH_i_	0.98	0.96	0.94	0.97	0.96	0.98	0.96

N.D. not detected.

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
