# Peer review of "Starch Retrogradation in Rice Cake: Influences of Sucrose Stearate and Glycerol"

_foods, 2020, doi:10.3390/foods9121737_

Round 1
Reviewer 1 Report
Dear Authors,
This is a very interesting study on the retrogradation of starch in rice cake affected by two additives. The introduction to the study and methods are all very good. But I think the results and discussion section need to be improved a lot.
- Title. "Macro-, micro- and molecular level studies ..." implies that many studies are reported here. The title could be something like "Study on starch retrogradation in rice cake at macro-, micro-, and molecular levels: effect of surfactant and plasticizer"
- There is much redundant information used again and again in this section, for example, about the glycerol's role as plasticizer and anti-plasticizer. Sometimes, different explanations were given regarding this mechanism. Line 145-147 "providing space for molecules to move within the polymer structure" & "block the aggregation of starch chains" make little sense or sound nonprofessional. Line 215 "depending on each concentration..." is not a good expression. Line 215-219 "In a fresh ...". I like these two sentences. These provide sufficient and valid explanations to the phenomenon. Anyway, this section needs a lot of work to make it concise and consistent.
- Another big issue is the lack of statistical analysis data. It is stated in the methods that means were compared along with ANOVA. However, none of the figures and tables are showing any statistical analysis data and labeling of the means comparison.
- Hardness. What is the unit of hardness here? The SI unit of hardness is N/mm². I would suggest that you use a letter to designate hardness, so that you wouldn't run into the problem of mixing words and letters in sentences in Line 124-126. It is very confusing here.
- Line 150. Rice flour is not going to form gluten.
- Line 156. "formation of amylose-lipid complexes was also confirmed in our XRD results". Amylose-lipid complexes might have formed, but your XRD results cannot confirmed this, because you said that "this seems to be due to partial recrystallization during the dehydration process using absolute ethanol" in line 170.
- About XRD data. Since you dried the rice cake samples using ethanol, you could not preserve the crystalline structure in rice cake so your XRD patterns were not those of the original rice cake but were manipulated by ethanol. This is a big drawback of the study. Ethanol can cause polymorphic transition, therefore, other drying and grounding methods should have been used, such as freeze drying and cryomilling.
- Line 227-228 "B-type crystalline structure, which contains 36 water molecules per 12 glucose residues". This was known from the modeling using XRD crystallography, but is still uncertain. But even if this was a correct B-type starch model, I don't think the water molecules immobilized this way are significant in affecting the amount of free water and bound water.
- Some minor issues: Line 54 the expression is very vague. Line 57 delete "a". Line 61 "in terms of" should be changed to "on the". Line 79 "referred to as". Line 86, what is "chart speed"? Line 87, "height of the first peak". Here is where you want to introduce the unit of hardness. Line 138 "slowed" - "decreased" or "slowed down". Line 139 "final hardness" was it final or should it be "maximum hardness"? Line 167 "divided into" - "classified" or "categorized". Line 196, "like hardness, ..." implies that hardness was inhibiting the recrystallization. And it should be recrystallization of starch in rice cake. Line 197 "the effect materialized at different times indicating that". This sentence is confusing.
- Numbers and units should be separated. "5 °C". and "°" should be consistent.
Author Response
Response to reviewer 1
This is a very interesting study on the retrogradation of starch in rice cake affected by two additives. The introduction to the study and methods are all very good. But I think the results and discussion section need to be improved a lot.
- "Macro-, micro- and molecular level studies ..." implies that many studies are reported here. The title could be something like "Study on starch retrogradation in rice cake at macro-, micro-, and molecular levels: effect of surfactant and plasticizer"
Ans) Thank you for your suggestion. Other reviewers also suggested a proper title, so we changed the title to ‘Starch retrogradation in rice cake: influences of sucrose stearate and glycerol’. Please see line 2-3 in the revised manuscript.
- There is much redundant information used again and again in this section, for example, about the glycerol's role as plasticizer and anti-plasticizer. Sometimes, different explanations were given regarding this mechanism.
Line 145-147 "providing space for molecules to move within the polymer structure" & "block the aggregation of starch chains" make little sense or sound nonprofessional.
Ans) Thank you for your suggestion. We used the references 11 and 18, which explained the anti-plasticizer role of glycerol in bread. We rephrased these sentences to make it sense. Please see line 152-154 in the revised manuscript.
Line 215 "depending on each concentration..." is not a good expression. Line 215-219 "In a fresh ...". I like these two sentences. These provide sufficient and valid explanations to the phenomenon. Anyway, this section needs a lot of work to make it concise and consistent.
Ans) Thank you for your suggestion. We rephrased these sentences to make it concise and consistent. Please see line 223-229 in the revised manuscript.
- Another big issue is the lack of statistical analysis data. It is stated in the methods that means were compared along with ANOVA. However, none of the figures and tables are showing any statistical analysis data and labeling of the means comparison.
Ans) Thanks for the suggestion. Although we conducted the Duncan's multiple range test for significant differences between mean values, it was difficult to express and label the mean comparisons in the figures because there were too many data in one graph. As reviewer insisted, mean comparison did not appear in the figures and table, we removed the mean comparison part and rewrite the statistical analysis. Please see lines 134-137 in the revised manuscript.
- What is the unit of hardness here? The SI unit of hardness is N/mm². I would suggest that you use a letter to designate hardness, so that you wouldn't run into the problem of mixing words and letters in sentences in Line 124-126. It is very confusing here.
Ans) Thanks for the suggestion. In this study, the unit of hardness is Newton (N). The cross-sectional area of rice cake samples and probe were set constantly for the hardness measurement. Therefore, in our study, the force to press the rice cake (hardness) is expressed in Newton without considering the cross-sectional area (mm²). We added the designation of hardness and rewrite as you suggested in the revised manuscript. Please see line 88 and 125-130 in the revised manuscript.
- Line 150. Rice flour is not going to form gluten.
Ans) Thank you for your suggestion. We wanted to propose that glycerol retard firmness in other starchy foods, such as bread as well as rice cakes. High concentration of glycerol increased firmness by plasticizing and strengthen the starch and gluten. We changed it accordingly. Please see line 155-157 in the revised manuscript.
- Line 156. "formation of amylose-lipid complexes was also confirmed in our XRD results". Amylose-lipid complexes might have formed, but your XRD results cannot confirmed this, because you said that "this seems to be due to partial recrystallization during the dehydration process using absolute ethanol" in line 170
Ans) Thank you for your suggestion. V-type polymorph can be obtained from amylose-lipid complex and partial recrystallization of starch. We change it accordingly. Please see line 164-167 and 176-178 in the revised manuscript.
- About XRD data. Since you dried the rice cake samples using ethanol, you could not preserve the crystalline structure in rice cake so your XRD patterns were not those of the original rice cake but were manipulated by ethanol. This is a big drawback of the study. Ethanol can cause polymorphic transition, therefore, other drying and grounding methods should have been used, such as freeze drying and cryomilling.
Ans) Thank you for your suggestion. In spite of polymorphic transition by ethanol, we used ethanol to prevent further retrogradation during other drying progress, which takes very long time. We believe that quick removal of water is important to minimize the further retrogradation of rice cake. Moreover, we treated all samples with same process and we can compare the samples. Additionally, the optimal conditions to manifesting the starch with V type crystal was ethanol treatment at 50°C for 120 min (Liu, Xie and Shi, Starch-Starke 68, 683-690, 2016). In our case, we used very short time of ethanol treatment with trace amounts of ethanol and dried at 40°C. Consequently, we thought that the effect of ethanol on polymorphic transition of starch crystal type is not significant.
- Line 227-228 "B-type crystalline structure, which contains 36 water molecules per 12 glucose residues". This was known from the modeling using XRD crystallography, but is still uncertain. But even if this was a correct B-type starch model, I don't think the water molecules immobilized this way are significant in affecting the amount of free water and bound water.
Ans) Thank you for your suggestion. The results were considered based on the most probable logics known to date for the XRD crystallography of starch. It is thought that water molecules could be considered bound water as they are contained inside the crystal lattice. Eventually this can affect the amount of freezable water. It has been reported that the amount of free water decreased as retrogradation progresses, leading to an increase in the glass transition temperature (Baik et al, Journal of Agricultural and Food Chemistry 45(11), p4242-4248, 1997). Consequently, as cited in references, these concepts are not our own proposal but hypothesis insisted by many researchers.
- Some minor issues:
Line 54 the expression is very vague.
Ans) Thank you for your suggestion. We rewrite this paragraph for better understanding. Please see line 52-54 in the revised manuscript.
Line 57 delete "a".
Ans) Thank you for your suggestion. We correct it as you suggested. Please see line 57 in the revised manuscript.
Line 61 "in terms of" should be changed to "on the".
Ans) Thank you for your suggestion. We correct it as you suggested. Please see line 61 in the revised manuscript.
Line 79 "referred to as".
Ans) Thank you for your suggestion. We correct it as you suggested. Please see line 80 in the revised manuscript.
Line 86, what is "chart speed"?
Ans) Thank you for your suggestion. Chart speed is the speed at which changes in force are imaged and passing by on chart paper. The texture analyzer we used in this study is an analog type which is not connected to a computer. In this case, it is common to count area squares or even measure a length of traces in chart recorder paper to quantify data. The slower the chart speed, the sharper the image will be. The faster the speed, the larger and the less detailed image you will get. We determined this chart speed through preliminary experiments.
Line 87, "height of the first peak".
Ans) Thank you for your suggestion. We correct it as you suggested. Please see line 88 in the revised manuscript.
Here is where you want to introduce the unit of hardness.
Ans) Thank you for your suggestion. We correct it as you suggested. Please see line 88 in the revised manuscript
Line 138 "slowed" - "decreased" or "slowed down".
Ans) Thank you for your suggestion. We correct it as you suggested. Please see line 145 in the revised manuscript
Line 139 "final hardness" was it final or should it be "maximum hardness"?
Ans) Thank you for your suggestion. We correct it as you suggested. Please see line 146 in the revised manuscript
Line 167 "divided into" - "classified" or "categorized".
Ans) Thank you for your suggestion. We correct it as you suggested. Please see line 174 in the revised manuscript
Line 196, "like hardness, ..." implies that hardness was inhibiting the recrystallization. And it should be recrystallization of starch in rice cake.
Ans) Thank you for your suggestion. We correct it as you suggested. Please see line 203-204 in the revised manuscript
Line 197 "the effect materialized at different times indicating that". This sentence is confusing.
Ans) Thank you for your suggestion. We rewrite this section for better understanding. Please See line 204-206 in revised manuscript.
- Numbers and units should be separated. "5 °C". and "°" should be consistent.
Ans) Thank you for your suggestion. We correct it as you suggested.

Reviewer 2 Report
Review for manuscript ID foods-974541
This manuscript aims to contain some information on starch retrogradation as affected by surfactant and plasticizer. Throughout the experiments waxy starch from rice has been used.
When reading the manuscript I had several difficulties:
- Wild type starch from rice has not been included. There is some evidence that waxy starch structurally differs from wild type starch and the results obtained might be valid only for that type of starch.
- Reserve starch usually contains both amylopectin and amylose but in waxy starch the content of amylose tends to be reduced. In the manuscript, quite frequently 'starch molecules' are mentioned (e.g. line 33; line 181; line 208; and line 216) which actually do not exist.
- The text is sometimes not completely clear (e.g. line 47-54; line 77f; line 120-126).
- The authors mention repeatedly the Avrami equations (line 63 and others). They do, however, not explain this term. Presumably, that goes back to a paper published in 1939 by M. Avrami and describing, in a very general way, kinetics of the phase transition. It seems to me that local events are more important for retrogradation taking place only at a part of the macromolecule(s).
Author Response
Response to reviewer 2
This manuscript aims to contain some information on starch retrogradation as affected by surfactant and plasticizer. Throughout the experiments waxy starch from rice has been used.
When reading the manuscript I had several difficulties:
Ans) Thank you for your suggestion. We did not use waxy rice flour but used normal rice.
- Wild type starch from rice has not been included. There is some evidence that waxy starch structurally differs from wild type starch and the results obtained might be valid only for that type of starch. Reserve starch usually contains both amylopectin and amylose but in waxy starch the content of amylose tends to be reduced.
Ans) Thank you for your suggestion. In this study, we used wild type (normal) rice flour for preparation of rice cake. There was mistyped in preparation section and we changed accordingly. Please see line 66 and 71 in the revised manuscript.
- In the manuscript, quite frequently 'starch molecules' are mentioned (e.g. line 33; line 181; line 208; and line 216) which actually do not exist.
line 33
Ans) Thank you for your suggestion. We rewrite it accordingly. Please see line 33 in revised manuscript.
line 181
Ans) Thank you for your suggestion. We rewrite it accordingly. Please see line 186 and 188 in revised manuscript.
line 208
Ans) Thank you for your suggestion. We rewrite it accordingly. Please see line 216 in revised manuscript
line 216
Ans) Thank you for your suggestion. We rewrite it accordingly. Please see line 224 in revised manuscript
- The text is sometimes not completely clear
line 47-54;
Ans) Thank you for your suggestion. We rewrite this section for better understanding. Please see line 47-54 in revised manuscript.
line 77f;
Ans) Thank you for your suggestion. We rewrite this section for better understanding. Please see line 75-78 in revised manuscript.
line 120-126.
Ans) Thank you for your suggestion. We rewrite this section for better understanding. Please see line 120-132.
- The authors mention repeatedly the Avrami equations (line 63 and others). They do, however, not explain this term. Presumably, that goes back to a paper published in 1939 by M. Avrami and describing, in a very general way, kinetics of the phase transition. It seems to me that local events are more important for retrogradation taking place only at a part of the macromolecule(s).
Ans) Thank you for your suggestion. Avrami equation has been used in analysis of retrogradation kinetics of starch materials by many researchers. They applied Avrami equation using firmness, DSC results, XRD results and NMR results etc. for retrogradation kinetics. Consequently, Avrami equation is known to general method to analyzed starch retrogradation kinetics. However, as you suggested. Avrami equation is not always adequate to analyze starch retrogradation and there is a limit to discussing the kinetics in the local event. Therefore, we applied both Avrami equation and non-linear regression to the macro, micro and molecular level retrogradation properties and concluded that non-linear regression is more appropriate to analyze some of retrogradation properties.

Reviewer 3 Report
The article is devoted to physico-chemical investigation of starch retrogradation in traditional rice cake and the influence of sucrose FA ester and glycerol on this process. The article is well written and the results obtained are well illustrated by clear graphical outputs. There are some minor points to revision:
- Title and text: I think that it is not necessary to stress on three levels of investigation, because, for example, DSC is difficultly to define as micro- or macro- method. Also, it is better to define the compounds chosen as modifiers of starch retrogradation, i.e. "Starch retrogradation in rice cake: an influence of sucrose stearate and glycerol".
- Abstract: Could authors recommend using sucrose stearate and/or glycerol to improve technology of rice cake production based on this study?
- I recommend to add macroscopic image of experimental rice cake because many readers of Foods are not familiar with this traditional Korean meal.
- Is it possible to use both sucrose stearate and glycerol together in rice cake? Can these substances enhance or weaken each other's action?
- Discussion: Could starch retrogradation be beneficial as a process leading to the formation of resistant starch type 3 and thereby push rice cakes towards diet food? Could the use of surfactants and/or plasticizers be aimed at regulating starch retrogradation rather than completely inhibiting it?
Author Response
Response to reviewer 3
The article is devoted to physico-chemical investigation of starch retrogradation in traditional rice cake and the influence of sucrose FA ester and glycerol on this process. The article is well written and the results obtained are well illustrated by clear graphical outputs. There are some minor points to revision:
- Title and text: I think that it is not necessary to stress on three levels of investigation, because, for example, DSC is difficultly to define as micro- or macro- method. Also, it is better to define the compounds chosen as modifiers of starch retrogradation, i.e. "Starch retrogradation in rice cake: an influence of sucrose stearate and glycerol".
Ans) Thank you for your suggestion. Other reviewers also suggested a proper title, so we considered and changed the title of this study to ‘Starch retrogradation in rice cake: influences of sucrose stearate and glycerol’. Please see line 2-3 in the revised manuscript.
- Abstract: Could authors recommend using sucrose stearate and/or glycerol to improve technology of rice cake production based on this study?
Ans) Thank you for your suggestion. The results of this research suggests that sucrose stearate and glycerol are effective in inhibiting retrogradation of rice cakes and will improve the industrial problems at hand. We added this suggestion in the revised manuscript. Please see lines 21-23 in the revised manuscript.
- I recommend to add macroscopic image of experimental rice cake because many readers of Foods are not familiar with this traditional Korean meal.
Ans) Thank you for your suggestion. Macroscopic images of traditional Korean rice cakes are included in graphical abstract. Please see the graphical abstract.
- Is it possible to use both sucrose stearate and glycerol together in rice cake? Can these substances enhance or weaken each other's action?
Ans) Thank you for your suggestion. We think that there is no problem in using both sucrose stearate and glycerol in rice cake unless they have the regulation. The action, whether to enhance or weaken on starch retrogradation will depend on the amount of each additive added. Thanks for the great suggestion that gives future research direction.
- Discussion: Could starch retrogradation be beneficial as a process leading to the formation of resistant starch type 3 and thereby push rice cakes towards diet food?
Ans) Thank you for your suggestion. The retrograded starch can cause the increasing of resistant starch type 3 content in rice cakes. However, rice cakes as a commercial product must have a soft and sticky texture, and rice cakes with hard texture are not selected by consumers. Therefore, further research would be required on how to increase the resistant starch and maintain its texture at the same time.
- Could the use of surfactants and/or plasticizers be aimed at regulating starch retrogradation rather than completely inhibiting it?
Ans) Thank you for your suggestion. In this study, we reported that the retrogradation-retardation effect was different depending on the added concentration of glycerol and sucrose stearate. Therefore, we believe that the retrogradation process of starch can be controlled by selecting the appropriate additives with adequate concentration.

Reviewer 4 Report
The article " Macro-, micro-and molecular level studies on starch retrogradation of rice cake: effects of surfactant and plasticizer" is interesting and describes significant research. The authors conducted interesting research and provided practical conclusions. My comment is only about statistics. There is no "standard deviation" marked in some figures. The graph in the graph (Fig.3a) is also unclear.
My suggestion for the future - Retrogradation of starch makes it less susceptible to the action of amylolytic enzymes. (Resistant starch is formed). Does slowing down the retrogradation process by adding glcerol and sucrose stearic acid ester affect the content of resistant starch? I think it is worth the authors to answer this question in the next article.
Author Response
Response to reviewer 4
The article " Macro-, micro-and molecular level studies on starch retrogradation of rice cake: effects of surfactant and plasticizer" is interesting and describes significant research. The authors conducted interesting research and provided practical conclusions.
- My comment is only about statistics. There is no "standard deviation" marked in some figures.
Ans) Thank you for your suggestion. There is a standard deviation in all graphs, but the value is too small to be noticeable. To clarify the standard deviation, we reduced the size of symbol in the graph. Please see Figures in revised manuscript.
- The graph in the graph (Fig.3a) is also unclear.
Ans) Thank you for your suggestion. We readjusted the y-axis of the graph to clarify it. Please see Fig. 3a in the revised manuscript.
- My suggestion for the future - Retrogradation of starch makes it less susceptible to the action of amylolytic enzymes. (Resistant starch is formed). Does slowing down the retrogradation process by adding glycerol and sucrose stearic acid ester affect the content of resistant starch? I think it is worth the authors to answer this question in the next article.
Ans) Thank you for your suggestion. In this study, it was confirmed that the addition of glycerol and sucrose stearic acid ester retard the starch recrystallization. Therefore, we think they may have less amount of resistant starch. The relationship between retrogradation and digestibility of starch under addition of glycerol and sucrose stearic acid ester needs to be investigated in future studies. Thanks for the great suggestion that gives future research direction.

Round 2
Reviewer 2 Report
During revision, some improvement of the submitted manuscript has been achieved. Still, I think some abbreviations in the Abstract should be omitted (e.g. XRD line 4). In the copy that was accessible to me there was a deletion of the text at the end of the Abstract.
Author Response
Thank you for your suggestion. We omitted the abbreviation and used the full name in the Abstract. We checked corrected the last sentence of Abstract. We marked it in red in the revised manuscript.
